# Exploring the Potential of Cytomegalovirus-Based Vectors: A Review

**DOI:** 10.3390/v15102043

**Published:** 2023-10-02

**Authors:** Janine Zeng, Dabbu Kumar Jaijyan, Shaomin Yang, Shaokai Pei, Qiyi Tang, Hua Zhu

**Affiliations:** 1Department of Microbiology and Molecular Genetics, New Jersey Medical School, Rutgers University, 225 Warren Street, Newark, NJ 070101, USA; janine.zeng.6@gmail.com (J.Z.); dkj28@njms.rutgers.edu (D.K.J.); sp2326@gsbs.rutgers.edu (S.P.); 2Department of Pain Medicine and Shenzhen Municipal Key Laboratory for Pain Medicine, Huazhong University of Science and Technology Union Shenzhen Hospital, Shenzhen 518060, China; shaominyang@stu2020.jnu.edu.cn; 3Department of Microbiology, Howard University College of Medicine, 520 W Street NW, Washington, DC 20059, USA

**Keywords:** HCMV, cytomegalovirus, CMV, viral vector, gene therapy, vaccine vector, rhesus CMV, murine CMV, bacterial artificial chromosomes

## Abstract

Viral vectors have emerged as powerful tools for delivering and expressing foreign genes, playing a pivotal role in gene therapy. Among these vectors, cytomegalovirus (CMV) stands out as a promising viral vector due to its distinctive attributes including large packaging capacity, ability to achieve superinfection, broad host range, capacity to induce CD8+ T cell responses, lack of integration into the host genome, and other qualities that make it an appealing vector candidate. Engineered attenuated CMV strains such as Towne and AD169 that have a ~15 kb genomic DNA deletion caused by virus passage guarantee human safety. CMV’s large genome enables the efficient incorporation of substantial foreign genes as demonstrated by CMV vector-based therapies for SIV, tuberculosis, cancer, malaria, aging, COVID-19, and more. CMV is capable of reinfecting hosts regardless of prior infection or immunity, making it highly suitable for multiple vector administrations. In addition to its broad cellular tropism and sustained high-level gene expression, CMV triggers robust, virus-specific CD8^+^ T cell responses, offering a significant advantage as a vaccine vector. To date, successful development and testing of murine CMV (MCMV) and rhesus CMV (RhCMV) vectors in animal models have demonstrated the efficacy of CMV-based vectors. These investigations have explored the potential of CMV vectors for vaccines against HIV, cancer, tuberculosis, malaria, and other infectious pathogens, as well as for other gene therapy applications. Moreover, the generation of single-cycle replication CMV vectors, produced by deleting essential genes, ensures robust safety in an immunocompromised population. The results of these studies emphasize CMV’s effectiveness as a gene delivery vehicle and shed light on the future applications of a CMV vector. While challenges such as production complexities and storage limitations need to be addressed, ongoing efforts to bridge the gap between animal models and human translation continue to fuel the optimism surrounding CMV-based vectors. This review will outline the properties of CMV vectors and discuss their future applications as well as possible limitations.

## 1. Introduction

Viral vectors are engineered versions of a virus that are precisely manipulated to effectively express foreign genes into host cells by taking advantage of the inherent gene transfer capability of viruses. Cytomegalovirus (CMV), a member of the herpesvirus family with a large double-stranded genome, has emerged as a novel and promising viral vector candidate for gene delivery due to its distinctive attributes and associated advantages. The complete CMV genome has been cloned into a plasmid, forming a bacterial artificial chromosome (BAC) that can be engineered to express foreign genes [1,2]. CMV holds a large packaging capacity that enables the accommodation of multiple foreign genes within its genome. Moreover, CMV can evade pre-existing immunity to induce strong T-cell responses. Numerous studies have found CMV-based vectors to be a versatile vehicle that can express genes related to malaria [3], TB [4,5], cancer [6,7], HIV [8,9], COVID-19, immunocontraception, aging, influenza, and other important applications. Crucially, CMV does not integrate into the host genome. Single-cycle CMV vectors generated by the deletion of essential genes can be utilized in an immunocompromised population due to robust inhibition of virus replication [10,11]. They are widely recognized as highly efficient gene transfer agents and find applications across diverse domains, ranging from genetic research to therapeutic interventions [12].

CMV has a large genome of approximately 230 to 240 kbp in length. It is highly species-specific, meaning that the virus is tightly restricted to its natural host. For instance, human cytomegalovirus (HCMV) only infects humans and rhesus CMV (RhCMV) only infects rhesus macaques or evolutionarily similar species. HCMV exhibits unique behavior by establishing lifelong latent infection that often remains asymptomatic. In immunocompromised individuals, HCMV reactivates and may cause severe and potentially life-threatening diseases, posing a major threat to the host’s health [13]. This has spurred extensive research into the virus’ infection, pathogenicity, immunity, and biology. Specific aspects of CMV conveniently resolve common limitations of other viral vector candidates, including challenges related to immune interference and packaging capacity [14]. Nonetheless, CMV also has several limitations that warrant consideration for its potential as a viral vector.

Several other herpesviruses, such as herpes simplex virus (HSV-1) [15], have also been harnessed as viral vectors. HSV-1 can infect epithelial and brain cells, making it a valuable tool for delivering treatments for neurological diseases. HSV-1 establishes latency in sensory ganglia [16] and can persist throughout the host’s lifetime. HSV-1 has been remarkably successful in delivering target genes to nervous system cells [17,18]. A replication-defective HSV-1 vector has ensured excellent safety in animals treated with it [19,20]. The versatility of the HSV-1 vector is further illustrated by its success in delivering treatments for bacterial diseases [21], HIV [22], cancer [23], Parkinson’s disease [24], Alzheimer’s disease [25], chronic pain [26], and other debilitating conditions.

In gene therapy, adenovirus, lentivirus, and adeno-associated virus (AAV) vectors have played crucial roles [27,28]. Adenovirus (Ad) vectors, for instance, have been widely used in gene therapies, including applications in cancer treatment and COVID-19 therapies [29]. Notably, the FDA recently approved an AAV-based gene therapy for Duchenne muscular dystrophy [30], reaching a significant milestone in gene therapy research. Table 1 provides a comprehensive review of CMV vector-based gene therapies and other viral vector-based gene therapies. According to global clinical trial reports, there are currently over 1000 ongoing clinical trials that harness viral vectors as a platform for gene therapy (http://www.abedia.com/wiley/vectors.php (accessed on 20 August 2023)). These clinical trials include ~575 adenovirus-based clinical trials, ~250 AAV-based clinical trials, and ~315 lentivirus-based clinical trials [31]. The current landscape of gene therapy research suggests that viral vectors represent an exceptional foundation for advancing gene therapy applications. This review aims to concisely summarize ongoing research on CMV-based vectors and discuss future applications of CMV-vectored gene therapy, with a particular emphasis on its role in vaccine vector developments.

## 2. Properties of Cytomegalovirus Vectors

CMV possesses unique assets that make it an attractive platform for gene delivery. It has infected a large percentage of the human population, totaling around 60% to 70% of adults in industrialized countries and almost 100% in developing countries [13]. Because CMV has been introduced to a large portion of the population and studied extensively, its pathogenicity and pathophysiology are well-known. Thus, a CMV-based vector will not cause additional unknown disease burdens within its host. Moreover, CMV has emerged as an ideal vector for viral vector-based gene therapy and vaccines because it has a large packaging capacity, possesses a broad cell tropism, can achieve secondary infection despite previous immunity, has high productivity in cell cultures known to have a stable viral genome, is safe and well-tolerated in humans, and induces long-term memory inflation (Figure 1).

CMV’s favorable safety profile is of significant importance. Despite the virus’ potential to cause severe diseases in immunocompromised individuals or birth defects during congenital infection, CMV’s safety concern has been largely reduced. This is due to the use of attenuated CMV virus strains that lack pathogenicity, such as laboratory-adapted strains AD169 and Towne, as foundations to create viral vectors. Clinical evaluations of these virus strains have found these attenuated strains to be safe and well-tolerated, ensuring safe translation to humans [32,33,34]. Further investigation of the live attenuated virus AD169 in clinical evaluation as a live attenuated virus has demonstrated that the virus does not cause any active infection in human subjects. Even when administered at doses as high as 300,000 pfu, AD169 did not elicit any pathological symptoms or result in viral shedding.

Additionally, a distinguishing feature of CMV lies in its substantial packaging capacity. CMV’s large genome of approximately 230 kbp in size, along with its multiple dispensable genes, allows a large number of foreign genes to be inserted into the CMV genome without compromising essential viral replication in vitro [35]. In comparison to viruses that have a smaller genome, CMV vectors deliver significantly larger pieces of DNA to targeted host cells. The laboratory-adapted strains Towne and AD169 lack a ~13 and a 15-kb segment, respectively, in the UL/b′ region, enabling the accommodation of a considerable number of foreign genes [36]. While these strains cannot replicate in vivo, they grow well in tissue culture, making them well-suited as foundations for viral vectors. Furthermore, CMV does not integrate into the genome of host cells, therefore alleviating concerns of genomic instability.

CMV vectors also resolve the key issue of virus immunity. Commonly, adaptive immune responses to viruses block secondary infections by the same or similar pathogens, preventing viruses from reinfecting a host. However, CMV is a notable exception as it is capable of achieving secondary infection, or superinfection, despite previous CMV exposure and immunity [37]. A study using rhesus macaques and RhCMV has revealed that CMV achieves this by evading CD8+ T cell immune responses through major histocompatibility complex class I (MHC-1) interference [38]. Notably, US2-11 glycoproteins play a major role in promoting evasion of CD8+ T cells in vivo, contributing to infections in individuals who should be CMV-resistant [38]. Although this ability to persistently reinfect hosts hinders the development of a CMV vaccine per se, it offers an avenue for CMV vectors to perpetually deliver genes to hosts regardless of prior immunity.

Within its host, CMV infects a wide range of cell types both in vivo and in vitro, including but not limited to epithelial cells, connective tissue cells, hepatocytes, and vascular endothelial cells, demonstrating a broad cell tropism [39]. This diverse infectivity translates to CMV’s capacity to provoke disease in multiple organs and tissues [40]. Noteworthy findings indicate that the HCMV gH/gL core complexes facilitate entry into the cell and adapt based on the cell type, with alternate gH/gL complexes influencing cell tropism, the entry pathway, and virus dissemination [41]. Due to its broad cell tropism, CMV is a versatile vector suitable for a wide range of applications. For instance, because CMV can naturally infect hematopoietic progenitor cells, it is a promising candidate for gene transfer into hematopoietic cells [42].

CMV promotes high and consistent gene expression within its host. Expression of CMV genes is temporally regulated, designated as immediate-early (IE), early (E), and late (L) gene expression phases [43]. HCMV’s major IE (MIE) genes include UL123 (IE1) and UL122 (IE2) which encode crucial proteins for regulating viral gene expression [44,45,46]. IE1 is imperative for replication at a low multiplicity of infection (MOI), while IE2 is required for early viral gene expression and viral replication [47,48,49,50]. Promoters derived from the enhancer region of the CMV IE promoter strongly drive transgene expression in mammalian cells due to their robust and persistent gene expression and compatibility with multiple cell types [51,52]. Within gene therapy applications, CMV promoters have been extensively studied in clinical trials because of this characteristic [53,54]. Accordingly, the CMV MIE promoter is frequently included in CMV vectors to drive expression of the foreign gene they carry, provoking constitutive and stable gene expression.

Viral vectors are frequently utilized as viral platforms for delivering immunogens to hosts, demonstrating unique potential in vaccine development. In this aspect, CMV vectors have a unique potential to treat various pathogens, cancers, and other diseases. CMV periodically reactivates from latency, triggering a large, virus-specific CD8+ T cell response, a phenomenon known as “memory inflation” [55]. These CD8+ T cells play important roles in immune defense, eliminating pathogens and tumors by secreting cytotoxic effector cells that clear infection [56]. When CMV induces memory inflation, it maintains CMV-specific T cells that consist of approximately 5–10% of all T cells in a healthy host [57,58,59]. CMV-driven T cells do not show signs of exhaustion and are capable of migrating to almost all body tissues while maintaining functionality, stimulating a sustained and strong cellular immune response against infectious diseases and cancer [58,60]. Due to this unique ability to elicit memory inflation as well as sustain long-lived effector memory T cell responses, CMV holds a significant advantage as a vaccine vector. Furthermore, CMV vectors may be genetically tailored to deliver diverse CD8+ T cell responses that target each individual disease, maximizing the immune responses to infections [61]. 

The overall properties of a CMV-based vector, as discussed previously, are concisely summarized in Figure 1.

### 2.1. Single-Cycle Replication CMV Vectors

To ensure the safety of viral vectors in clinical applications, it is imperative to develop non-replicating CMV vectors, as these vectors cannot propagate in vivo and possess an excellent safety profile. Live-attenuated viruses, though promising, have been reported to reactivate even after immune suppression [62], raising significant concerns. As an alternative to live-attenuated viruses, single-cycle replication MCMV and RhCMV vectors have been engineered to effectively address these issues. Research shows that these vectors are non-pathogenic and do not spread to adjacent sites after initial infection [10,11], ensuring a satisfactory safety profile.

For instance, the deletion of the Rh10 gene in a RhCMV vector has reduced in vivo pathogenicity and replication in a study, yielding a live-attenuated RhCMV vector [63]. The removed Rh10 gene is known to encode the tegument protein pp150. Using this live attenuated RhCMV/SIV vector, a robust and protective immune response against SIV was generated [64], and importantly, RhCMV was unable to spread to new subjects. Another study demonstrated that the deletion of a *gL* gene from RhCMV can produce a single-cycle viral vector that induces a strong RhCMV-specific immune response [11].

Similarly, a single-cycle MCMV vector was generated by deleting an essential M94 gene (MCMV-Δ94). This vector, expressing ovalbumin (AVA), triggered strong cellular and humoral immune responses [10]. Additionally, the research found that a recombinant MULT-1MCMV generated by expressing a high-affinity immune receptor ligand is capable of inducing a strong, protective immune response against the target antigen expressed within the vector. The extensive investigation conducted on live-attenuated, single-cycle CMV vectors highlights their potential as a novel platform for expressing diverse antigens for vaccine and gene therapy applications while offering an enhanced safety profile.

### 2.2. CMV as a Self-Disseminating or Transmissible Vaccine Vector

The emergence of new viruses through zoonotic transmission presents a significant health concern, requiring preemptive measures to prevent potential epidemics or pandemics. To curb the transmission of viruses from animal to human (zoonosis), the vaccination of small animals that harbor viruses has been proposed. For this purpose, self-disseminating vectors have been considered as vaccine delivery vehicles. They could allow immunized animals to transmit the vaccine vector to non-immunized individuals via contagious or behavioral routes, reducing both costs and efforts [65]. However, identifying a transmissible vector that can escape pre-existing immunity and infect new hosts is a significant challenge. CMV stands out as a candidate for such as role, given its superinfection capability and the lack of protective immunity in previously immunized subjects [66]. Recent developments in the field have demonstrated the feasibility of using CMV as a transmissible vector for generating vaccines against a broad spectrum of pathogens.

In fact, a method has recently been devised for DNA viruses to enable the production of DNA virus vaccines for the vaccination of small animals. This method involves using a chemical approach to produce chemically attenuated DNA viruses for vaccine development [67]. Furthermore, a recent study has showcased the effectiveness of a CMV-vectored transmissible vaccine against a wide range of pathogens. The results of this study strongly suggest CMV’s utility as a valuable tool for developing vectored transmissible vaccines and as a promising strategy to prevent zoonotic transmission [68].

## 3. CMV-Based Vectors and Gene Therapy

There have been many potential gene therapies successfully completed in preclinical studies that are based on CMV vectors (Figure 2). Recently, CMV vectors have gained more attention in the field of vaccine development, especially as conventional vaccination strategies encounter difficulties against formidable diseases such as AIDS and cancer. CMV’s prominence as a vaccine vector is mainly attributed to its ability to induce robust antigen-specific CD8^+^ T cells. Using CMV as a vaccine or gene therapy vector has multiple advantages, one of which is that a CMV vector can be used as a dual vaccine that effectively prevents both natural CMV infection and the target antigen cloned within the CMV construct. With a focus on these characteristics, we will discuss the major applications of CMV vectors that have been investigated so far. The impressive achievements in numerous preclinical studies, which have utilized CMV vectors for gene therapies targeting conditions such as HIV, cancer, malaria, TB, aging, and more, emphasize the pivotal role of CMV vectors in the burgeoning field of gene therapy.

**Table 1 viruses-15-02043-t001:** List of CMV vector and other viral vector-based gene therapy.

Vaccine	Vector	Disease	Target Genes	References
TB vaccine	RhCMV	Tuberculosis	Rv3407, Ag85B, Ag85A, Rv2626, ESAT-6, Rv1733, RpFA, RpD and RpfC	[69]
SIV vaccine	RhCMV strain 68-1 (Attenuated)	Highly pathogenic SIV	Rev-tat-nef, gag, Env, Pol1 and Pol2	[63,64]
EBOLA vaccine	RhCMV strain 68-1	EBOV	EBOV GP	[70]
SIV vaccine	RhCMV strain 68-1	Highly pathogenic SIV	Rev-tat-nef, gag, Env, Pol1 and Pol2	[8,9,69]
Malaria vaccine	RhCMV strain 68-1	Malaria	CSP, SSP2, AMA1, MSP1	[71]
EBOLA vaccine	MCMV Smith strain containing m157 deletion	EBOV	ZEBOV NP	[72,73]
*Listeria monocytogenes* vaccine	Attenuated RAE-1γMCMV	*Listeria monocytogenes*	Liseriolysis O_91-99_	
Cancer Vaccine	MCMV containing FKBP-mediated destabilization of M79 gene	Tumor	HPV16 E749-57 epitope	[74]
TB vaccine	MCMV: deletion of MHC class I downregulators	Erdman strain Mtb	Mtb 85	
EBOLA vaccine	MCMV Smith strain with m17 deletion	ZEBOV	ZEBOV NP	[72,73]
RSV vaccine	MCMV K181 strain with m157 deletion	RSV	RSV M protein	[75,76,77]
Influenza vaccine	MCK2-repaired MCMV Smith strain	IAV PR8M	Peptide IVL_533-541_	[78]
HSV-1 vaccine	MCK2-repaired MCMV Smith strain	*HSV-1*	HSV-1 glycoprotein B_498-505_	[79]
Anti-aging gene therapy	MCMV Smith strain	*Agiing*	mTERT and mFST	[80]
Glioblastoma multiforme vaccine	HCMV strain TB40/E	Glioblastoma multiforme	HPV-16 concensus E6/E7 fusion protein	[55]
Cancer	HSV-1	Various cancer		[29]
Head and Neck cancer	Ad5	Head and Neck cancer	P53/Gendicine	[29]
EBOLA vaccine	VSV	ZEBOV		[29]
Ad26.COV2.S	Ad26	COVID19	JNJ-78436735	[29]
EBOLA vaccine	Ad26	ZEBOV	MVA-BN-Filo/Zabdeno	[29]
Sputnik V	Ad26 Ad5	COVID19	Sputnik V	[29]
Covishield	ChAD		AZD1222	[29]
Neurological disease	AAV9	Spinal Muscular atrophy	Onasemnogene Abeparvovec/ZOLGENSMA	[29]
Metabolic disease	AAV1	Familial hyperchylomicrenemia	Alipogene tiparvovec	[29]

### 3.1. Harnessing CMV Vectors for HIV Vaccination

The persistent global impact of human immunodeficiency virus (HIV) remains a critical concern, marked by an estimated 1.3 million new infections in 2022 and a staggering cumulative global death toll of approximately 40 million [81]. While combination antiretroviral therapy (ART) has been highly successful in prolonging the lifespan of the patient as well as limiting transmission, many HIV-infected individuals are either unaware of the infection or unable to access vital antiviral treatments [81]. Thus, it is imperative to develop a prophylactic HIV vaccine to significantly reduce transmission rates and enhance herd immunity [82,83]. However, because HIV is capable of evading host immunity and damaging immune cells, conventional vaccination strategies encounter formidable challenges in achieving long-term efficacy. In such circumstances, vaccine vectors have emerged as a promising approach to induce HIV-specific immune responses, particularly for a CMV-based vector that is capable of eliciting durable immune responses in an infected host [84,85].

A CMV vaccine vector has attained a certain level of success against simian immunodeficiency virus (SIV), a virus that is closely related to HIV and infects a large range of nonhuman primate hosts. An innovative study has developed a fibroblast-adapted rhesus cytomegalovirus (RhCMV) laboratory strain 68-1 as a delivery vector to express SIV antigens Gag, Rev/Nef/Tat and Env (RhCMV68-1/SIV) [8,9,86]. Impressively, 55% of rhesus macaques that were vaccinated with RhCMV68-1/SIV attained early infection control and sustained protection against the highly virulent SIVmac239 variant [8,86]. The vaccinated subjects consistently exhibited robust and persistent SIV-specific CD8+ T cell responses, indicating that RhCMV-vectored SIV vaccine triggered CD8+ T cells that played a pivotal role in the control of SIV proliferation [87]. Interestingly, the RhCMV68-1/SIV vaccine efficacy was shown to rely on unconventional, noncanonical CD8+ T cell responses [87,88]. Although RhCMV68-1/SIV was highly successful in preventing SIV infection, its protective ability diminished when administered to antiretroviral-treated rhesus macaques post-SIV infection, suggesting a narrow window of intervention for the viral vaccine to be effective [89]. Using RhCMV and SIV as a model, the success of this study implies that CMV holds promise as a vaccine vector for HIV. The construction of an effective CMV-vectored HIV vaccine may significantly relieve the global HIV epidemic.

### 3.2. Cancer Immunotherapy

As the pursuit of refined cancer treatment methodologies continues, cancer immunotherapy has emerged as a formidable alternative to conventional chemotherapy and radiation. In particular, cancer vaccines are considered a promising strategy that aims to use tumor antigen-specific immune responses to kill tumor cells [90]. Recombinant viral vectors have garnered significant attention as versatile vehicles for vaccine delivery, providing unique advantages as each virus has different properties that can be applied in specific therapeutical settings. Of notable interest is CMV, a promising cancer vaccine vector as CD8^+^ T cells promoted by CMV serve important roles in controlling tumor progression [6,7]. A noteworthy advantage of CMV-driven CD8^+^ T cells is that they do not become exhausted and could potentially generate enough of an immune response over time to control tumor growth in prophylactic or therapeutic settings [58,91]. So far, research exploring the possibility of a cancer vaccine using murine CMV (MCMV) as a gene delivery vehicle has undergone substantial advancements.

#### 3.2.1. Studies in Prostate Cancer

Multiple studies have examined the possibility of using MCMV as a conduit for gene delivery that functions as a cancer vaccine, propelling advances in prostate cancer immunotherapy. The first proof-of-concept study that demonstrated CMV’s efficacy as a cancer vaccine utilized MCMV-based vaccines expressing human prostate-specific antigen (PSA) for prostate cancer immunotherapy [92]. Within the transgenic adenocarcinoma of the mouse prostate (TRAMP) model, two MCMV-based vectors expressing either an H2-D^b^-restricted epitope PSA_65-73_ (MCMV/PSA_65-73_) or the full-length PSA gene (MCMV/PSA_FL_) were developed and tested. While both vectors induced similarly comparable levels of CD8 T cell responses that progressively escalated in the absence of tumor challenge, MCMV/PSA_65-73_ demonstrated the most tangible impact. When the animals were challenged with TRAMP-PSA tumor cells, MCMV/PSA_FL_ showed no increase in tumor-specific T cells and failed to slow tumor growth, while MCMV/PSA_65-73_ increased PSA-specific CD8 T cell responses within the host and subsequently slowed tumor growth. This pioneering study demonstrated that a CMV-based prostate cancer vaccine is capable of eliciting an effective anti-tumor immune response [92].

#### 3.2.2. Advances in Melanoma Therapeutics

In the pursuit of novel and effective avenues for cancer treatment, CMV has been explored as a cancer vaccine for melanoma multiple times, with numerous studies displaying the potential of CMV-based cancer vaccines in delaying tumor growth in melanoma models. A groundbreaking investigation led by the Hill laboratory featured the expression of unmodified melanoma antigen mouse tyrosinase-related protein 2 (TRP2) within MCMV (MCMV-TRP2), testing its effectiveness as both a prophylactical and therapeutical intervention in mice [93]. Prophylactic vaccination with MCMV-TRP2 yielded striking outcomes, resulting in the rejection of B16-F10 melanoma cells and the establishment of long-term protection against B16 melanoma challenge regardless of prior CMV infection. Therapeutic vaccination delayed tumor growth and prolonged mice’s lifespan. Intriguingly, the study uncovered that CD8+ T cells played a secondary role, while tumor-specific antibodies induced by MCMV-TRP2 were crucial for a strong and long-lasting anti-melanoma effect [93]. Subsequent research determined that the protective efficacy against melanoma in this model relied on the expression of FcγRI on macrophages, further revealing the complex interplay within the immunotherapeutic landscape [94].

The investigation of CMV-driven cancer vaccines against melanoma has manifested in diverse studies with different CMV-based vectors, each unveiling promising results. An innovative approach engineered a recombinant MCMV strain to express modified B16 melanoma antigen, gp100 (MCMV-gp100KGP), studying its potential in prophylactic and therapeutic settings within a B16 lung metastatic melanoma mice model [95]. This resulted in a robust and sustained gp100-specific CD8+ T cell response independent of prior CMV infection and immunity, and immunization with MCMV-gp100KGP resulted in tumor rejection in both prophylactic and therapeutic settings [95]. Another recombinant MCMV-based vaccine vector expressing only the modified CD8+ T lymphocyte epitope was tested, resulting in much slower tumor growth and increased lifespan after intratumoral administration, although the vaccine had limited success when administered systematically [96]. Further investigation suggests that the ability of MCMV to protect against melanoma relies significantly on boosting the activity of pre-existing tumor-specific CD8+ T lymphocytes and their synergistic action with tumor-associated macrophages [4,96,97].

### 3.3. CMV-Based Tuberculosis Vaccines

Tuberculosis (TB) persists as a significant global health challenge and is one of the leading causes of mortality worldwide with Mycobacterium tuberculosis (Mtb) as a single infectious agent. The widely used bacillus Calmette–Guérin (BCG) vaccine for tuberculosis affords limited protective capacity that diminishes with age [98]. Since Mtb has coexisted with humans for over 70,000 years, it has evolved to adapt unique immune evasion strategies that hinder efforts to develop effective solutions for the current TB epidemic [99]. Given CMV’s innate ability to consistently sustain antigen-specific CD8+ T cells and achieve secondary infection, it has recently been examined as a tuberculosis vaccine vector in both murine and rhesus macaque models.

A recombinant MCMV expressing Mtb Ag 85A (MCMV85A) was engineered using bacterial artificial chromosome (BAC) technology, the efficacy of which was assessed in BALB/c mice. The outcomes were remarkable, revealing the establishment of protective immunity against M. tuberculosis, a defense that endured for at least 24 weeks [5]. The MCMV85A vaccine generated a virus-specific adaptive response as well as a nonspecific protective effect against the M. tuberculosis challenge. Although this vaccine demonstrated efficacy against TB, the innate immunity of the host played a substantial role in MCMV85A’s protective capacity, which may restrict the vaccine’s long-term effectiveness [4,5].

Another study developed RhCMV vectors expressing Mtb antigen (Ag) (RhCMV/TB) to vaccinate rhesus macaques followed by challenges with the highly pathogenic Erdman strain [69]. The vaccination instigated the induction and sustenance of Mtb-specific CD4+ and CD8+ T cell responses with high effector differentiation, resulting in an impressive 68% reduction in Mtb infection and disease compared to unvaccinated controls. Additionally, out of 34 vaccinated rhesus macaques, 14 were cleared of TB disease, with 10 rhesus macaques demonstrating complete Mtb culture-negativity in all tissues. This achievement demonstrated a CMV-based vaccine’s capability in combating Mtb, provided immune effector responses can detect Mtb infection during its early stages [69].

In both studies, the CMV-vectored TB vaccines, MCMV85A and RhCMV/TB were shown to protect against TB largely because they sustained persistent *Mtb*-specific immune responses in the host. The inherent capability of CMV infections to trigger specific T cell responses indicates that CMV-vectored vaccines for humans will elicit similar effector-memory T cell responses, mirroring the potency of MCMV85A and RhCMV/TB against tuberculosis. Thus, the effectiveness of the MCMV- and RhCMV-based vaccine vector for TB implies the efficacy of a CMV-based vaccine for humans.

### 3.4. CMV Vaccine Vectors for Malaria

The relentless malaria epidemic has driven the endeavor for effective vaccines, particularly against the pre-erythrocytic stage. The sporozoite vaccines against the pre-erythrocytic stage of malaria are widely used, yet they lack a long-term effect due to their failure to maintain effector T-cell responses in the liver. Subsequently, this has encouraged the development of a new type of vaccine based on CMV, which can be viewed as a booster to extend protection against malaria. A study has found that *P. chabaudi* MSP-1 epitope B5 (MCMV-B5) was able to increase the population of highly differentiated Tem and B5-specific T cells, producing a formidable defense against malaria [3]. In addition, MCMV acted as an adjuvant to stimulate IFN-gamma, which increased CD8α+ dendritic cell numbers and caused increased IL-12 production.

Further illustrating the efficacy of a CMV vector, a study discovered that RhCMV was highly stable and successful in maintaining effector memory T cells in extra lymphoid tissues [9,71]. Notably, the strain 68-1 of RhCMV activated CD8+ T cells to recognize unconventional epitopes exclusively restricted by MHC-II and MHC-E. Four *Plasmodium knowlesi* (Pk) antigens (CSP, AMA1, SSP2/TRAP, MSP1c) were expressed in RhCMV 68–1 or Rh189-deleted 68–1. Furthermore, T-cell responses were successfully maintained in all rhesus macaques upon inoculation [71]. The study observed the delayed appearance of blood-stage parasites and significantly reduced parasite release from the liver, indicating that it is possible to control malaria with unconventional Tem-inducing RhCMV vectors and further improve current vaccines for malaria. With the success of RhCMV vectors as a model, it is likely that CMV can be effectively used as a vaccine vector for malaria in humans.

### 3.5. Influenza A and Coronavirus

Both severe acute respiratory syndrome coronavirus 2 (SARS-Cov-2) and influenza A virus (IAV) have caused global pandemics and led to high levels of morbidity and mortality. While certain effective vaccines have been developed to combat these viral infections and diseases, there exist a myriad of side effects such as fever and fatigue. Moreover, the vaccination regimens for both SARS-CoV-2 and IAV require a meticulous sequence of prime and boost vaccination protocols, and vaccines for IAV have limited efficacy ranging from 19% to 60% depending on the circumstance [100]. Within this specific field, viral vaccine vectors have been explored as an alternative avenue to help overcome these hurdles. Viral vectors have the significant advantage of eliminating the need for adjuvants and naturally inducing both cellular and humoral adaptive immune responses [101,102]. In this aspect, as a compelling candidate for vaccine delivery, CMV has been investigated as a potential vaccine vector to target both SARS-Cov-2 and IAV.

A recent study generated two recombinant MCMV vaccine vectors using BAC-based recombination: one expressing hemagglutinin of influenza A virus (MCMV^HA^), and the other the spike protein of SARS-CoV-2 (MCMV^S^) [103]. A single-dose administration of either vaccine vector to mice developed a potent neutralizing antibody response that strengthened over time, with MCMV^HA^-vaccinated mice also receiving immune protection after influenza infection. The immune protection offered by the vaccines was not reliant on CD8^+^ T cell responses, but rather protective B-cell memory responses triggered by the vaccines against the two pathogens. Through this study, it was discovered that MCMV vectors induce both long-term cellular immunity and immune protection against respiratory pathogens [103].

### 3.6. Gene Therapy for Aging

Aside from functioning as a vaccine vector, CMV has been applied to different areas of gene therapy, exerting its influence across a wide range of domains. Recent strides have witnessed CMV vectors applied to the field of gene therapy, specifically in the arena of aging. By using CMV as a safe and effective gene delivery vehicle to increase mouse lifespan, a recent study has demonstrated its potential to expand into the other broad applications of viral vectors.

Telomerase reverse transcriptase (TERT) is an important protein in telomerase activation that increases telomere length [104], with TERT-deficient animals possessing shorter telomeres [104]. Additionally, TERT-based complementary gene therapies have demonstrated that the administration of TERT can reduce aging in treated animals [105,106]. Similarly, follistatin (FST) overexpression increased muscle mass [107] and conversely, mice who were deficient in FST had few muscle fibers, showed skeletal defects, and died within a few hours of birth [108]. In a recent study, MCMV has been used as a delivery vector for anti-aging gene therapy by expressing TERT and FST proteins.

An engineered recombinant MCMV containing either mouse TERT or FST was developed using a BAC recombineering method and inoculated into 18-month-old mice via intraperitoneal or intranasal routes [109]. As a result, the protein levels of TERT and FST were significantly higher in the treated mice as opposed to their untreated or control counterparts. Rigorous analysis, including RT-PCR and serum analysis of the treated mice, demonstrated that CMV effectively delivered either TERT or FST to multiple mouse tissues. Importantly, the lifespan of MCMV-TERT-treated mice was increased by approximately 41% without undesirable side effects [80]. Equally striking, MCMV-FST-treated mice showed a 32.5% increase in lifespan over untreated mice. MCMV-TERT-treated mice had longer telomeres in different tissues compared to untreated mice, and the mice treated with MCMV-FST were found to be heavier with 33% more body weight than the untreated control. The anti-aging effects demonstrated by the two vectors extended beyond the physical domain, with mice treated with MCMV-TERT and MCMV-FST having not only better coordination and improved activity as compared with untreated mice but also increased glucose tolerance and better mitochondrial integrity. These outcomes demonstrate that MCMV had successfully delivered functional TERT and FST into injected mice, indicating that CMV is an exceptional viral vector to deliver target genes and may serve important roles in gene therapy applications in the future.

### 3.7. Ebola Virus (EBOV) Vaccine

EBOV, a member of the filovirus family, is an ssRNA, non-segmented, and enveloped negative RNA virus that causes severe hemorrhagic fever in humans. Human transmission occurs through contact with infected material such as blood, tissues, or patients, often resulting in multiple organ failure and death. EBOV is able to infect many cell types in the body, possessing a broad cell tropism. In preclinical studies, a CMV-based vector expressing EBOV antigen, specifically an epitope from the Zaire EBOV nucleoprotein, was tested. Immunization of mice with the MCMV vector expressing EBOV NP antigen generated robust CD+ T cell responses against EBOV NP, leading to complete protection against lethal EBOV challenge [72,73]. In another study, primates immunized with a RhCMV vector expressing EBOV glycoproteins developed protective immunity against EBOV [70]. This approach induced a strong IgG response and effectively shielded immunized animals from lethal EBOV challenges. The investigation of CMV viral vector vaccines against EBOV provides further evidence of the virus’s potential as a vaccine vector.

### 3.8. CMV Vector for Immunocontraception

Viral vectored immunocontraception (VVIC) involves administering genetically engineered viruses that are designed to induce infertility by stimulating an immune response against reproductive cells or essential reproductive antigens [110]. This birth control method disrupts normal reproductive function and leads to sterility or infertility. In one study, a wide range of self-antigens and synthetic polyepitope antigens were expressed within a MCMV vector to evaluate its potential as an immunocontraceptive vaccine. For example, different recombinant MCMV vectors were prepared using genetic engineering to express mouse bone morphogenic protein 15 (BMP15), murine zona pellucida 3 (mZP3), N-terminal ubiquitin tagged murine zona pellucida 3 (ZP3), and murine oviduct glycoprotein (OGP). Mice infected with MCMV expressing murine ZP3 and MCMV expressing ubiquitin tagged ZP3 were 100% infertile [110]. These results demonstrated MCMV’s success as a viral vector immunocontraceptive vaccine to induce sterility or infertility.

## 4. Discussion

Safety and efficacy are considered two of the most important factors in viral vector development. The benign nature of CMV’s attenuated strains combined with the efficacy of CMV-based vectors has ignited fervent interest within the scientific community. Additional properties of CMV such as its large packaging capacity, ability to achieve secondary infection, and broad cell tropism make CMV-based vectors compelling contenders for diverse applications in the field of gene therapy. However, CMV vectors possess several inherent limitations that may constrain their development and use.

A notable restriction arises in the context of vector development and production. CMV’s slower growth in tissue culture impairs its suitability for large-scale vector production [14]. Additionally, CMV’s strict species-specificity means the virus is only able to be studied within its limited natural hosts, making it difficult to conduct studies of its replication, pathogenesis, and other characteristics [111]. Moreover, CMV owns a large genome that is not fully understood, making it difficult to directly construct a recombinant CMV. Nevertheless, innovative tools such as Bacterial Artificial Chromosome (BAC) technology have paved a more efficient path. CMV genomes nestled within BAC constructs facilitate the manipulation of viral genetic makeup, fostering deeper comprehension and streamlined recombinant virus creation [35,112,113]. The use of BAC technology has made the identification of CMV gene functions, mutagenesis of the viral genome, and especially production of recombinant viruses simpler and more convenient [114]. The engineering of mutant CMV strains with BAC is more straightforward, as desired mutations can be easily achieved and confirmed inside the *E. coli* cell [113]. BAC also enables research on removing CMV genes that interfere with immune function, improving the efficacy of CMV-based vaccine vectors [115,116,117].

Live CMV vectors face the common challenge of prolonged storage within certain conditions. Live viral vaccine vectors are typically difficult to preserve and prone to instability, especially if exposed to high temperatures and multiple freeze-thaw cycles [118,119]. A recombinant HCMV vector (rHCMV-1) experienced large vector titer losses after being stored at 4 °C and undergoing a freeze-thaw cycle [120]. Further investigation revealed that the removal of NaCl, which decreased the ionic strength, and the incorporation of additives including sugars and polymers decreased viral titer losses caused by freeze-thaw cycles, while optimized solution pH, buffers, and sugar types protected rHCMV-1 titer losses against prolonged storage at 4 °C in liquid state [120]. However, maintaining the stability of live CMV-based vectors remains a problem for its long-term storage and widespread use and distribution.

CMV-based vectors have repeatedly demonstrated their potential as vaccines and gene delivery vehicles in multiple studies spanning murine and rhesus macaque models. While they have shown excellent efficacy in nonhuman models, it is unclear if the same results will be achieved when translated to humans. A major reason that CMV-based vaccine vectors are successful within non-human models is that CMV naturally induces memory inflation within its host [85]. Thus, RhCMV- and MCMV-based vectors that relied on sustained populations of antigen-specific CD8+ T cells to achieve effectiveness should theoretically provide highly similar results when translated to humans. Research indicates that HCMV orthologs of certain RhCMV genes, including UL128/130, UL146/UL147, and UL40, conserve the potent ability to program CD8+ T cell responses [121]. RhCMV and MCMV’s similarities to HCMV’s genomes, viral pathogenesis, and other functions also suggest that the outcomes of RhCMV and MCMV-based vectors in their respective non-human models can be effectively translated to humans [111,122,123]. However, the complex differences between RhCMV or MCMV and HCMV may still pose a significant challenge in translating their investigated efficacy to humans. For this reason, further investigation is crucial in understanding the potential translation of CMV vectors to humans. Nevertheless, studies with RhCMV and MCMV-based vectors have highlighted the importance of further refining HCMV vectors and researching how they may be developed to produce beneficial effects in humans.

Despite these challenges, CMV vectors have showcased their prowess across diverse ailments, affirming their potential in gene therapy and vaccine development. Multiple studies thus far have shown the success of CMV vectors in the treatment of diseases and other forms of gene therapy. Still, more research and clinical trials need to be conducted to better understand the mechanisms behind certain CMV characteristics and ensure efficient translation of CMV vectors to humans for potential clinical uses. The possible applications for viral vectors are broad, and for a CMV-based vector that holds unique advantages, a promising future lies ahead.

## Figures and Tables

**Figure 1 viruses-15-02043-f001:**
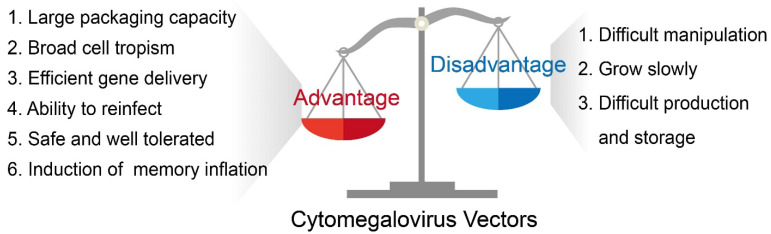
The basic advantages and disadvantages of a CMV-based vector.

**Figure 2 viruses-15-02043-f002:**
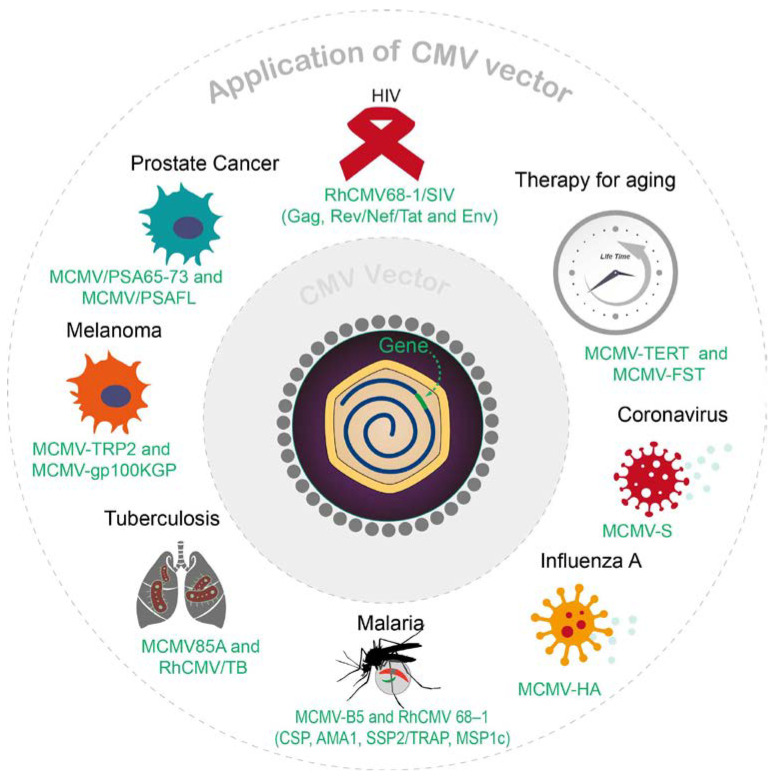
Overview of recent CMV vector applications.

## Data Availability

Not applicable.

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
