# Peer review of "Exploring the Potential of Cytomegalovirus-Based Vectors: A Review"

_viruses, 2023, doi:10.3390/v15102043_

Round 1

Reviewer 1 Report

Zeng et al. have prepared a overview of applications of cytomegaloviruses as vectors in medical therapy and vaccination.

Abstract:

The second sentence of the Abstract come across as a tautology in stating “stands out as a promising viral vector due to its distinctive attributes that make it an appealing vector candidate” such that it would be preferrable to simply start to list ”distinctive attributes” that the authors are referring to. The remainder of the Abstract does not provide very much in the way of concrete statements. Given that cytomegaloviruses are recognized pathogens that remain latent in a majority of the worldwide population, more must be said about how attenuation is achieved and how vectors can be used in an immune population.  Outright statements promising “human safety”, “incorporation of extensive foreign genes” and “effectiveness as a gene delivery vehicle” sound hollow without some tangible examples or insights to draw the reader in.

Introduction:

The term “engineered versions of a virus” is used without clearly defining the basic characteristics of a viral vector. Other herpesviruses that carry foreign genes have been made for medical and veterinary use and these should be included for full understanding. How is latency of these viruses managed?

Properties:

The authors make a bold claim that attenuated Towne and AD169 strains of CMV “cannot replicate in vivo, they grow well in tissue culture, making them well-suited as foundations for viral vectors.”  What evidence exists that the attenuation of these strains results in replication incompetence? The statement seems to be a stretch and most likely incorrect. When administered to CMV naïve human volunteers, Towne strain and AD169 strain, as well as Towne/Toledo chimera viruses were all immunogenic, interpreted as being the result of replication of attenuated virus because inactivated virus was not immunogenic. The section seeks to promote the point that CMV shows a “significant advantage as a vaccine vector” without clearly delineating the reasons why.

Figure 1 should be replaced by a figure or table that contracts CMV-based vectors with more commonly studied and FDA licensed viruses and vectors based on other herpesviruses as well as on other virus types. The current figure does little to assist the reader in gaining perspective.

Applications:

This section seeks to summarize what has been gained from studies of surrogate CMV models and so should be more accurately titled. All examples in Figure 2 employ non-human CMVs.  How closely related are these viruses in molecular terms? Can any enter and express vectored genes in human cells or actually replicate in human cells? If so, can they infect humans and be tested as replication competent or nonreplicating vectors? (It appears that RhCMV is actually grown in immortalized human fibroblasts and reports suggest that MCMV can be adapted to human RPE cells. Some overarching perspective is needed here.

Although examples are given in this section, the points are not synthesized or sufficiently integrated.

Very simple 8th grade level text.

Author Response

Comments and Suggestions for Authors

 Zeng et al. have prepared a overview of applications of cytomegaloviruses as vectors in medical therapy and vaccination.

Abstract:

The second sentence of the Abstract come across as a tautology in stating “stands out as a promising viral vector due to its distinctive attributes that make it an appealing vector candidate” such that it would be preferrable to simply start to list ”distinctive attributes” that the authors are referring to. The remainder of the Abstract does not provide very much in the way of concrete statements. Given that cytomegaloviruses are recognized pathogens that remain latent in a majority of the worldwide population, more must be said about how attenuation is achieved and how vectors can be used in an immune population.  Outright statements promising “human safety”, “incorporation of extensive foreign genes” and “effectiveness as a gene delivery vehicle” sound hollow without some tangible examples or insights to draw the reader in.

Response: We appreciate reviewer’s suggestion and constructive comments to improve the MS. We have made suggested changes in abstract in the revised manuscript (MS) (Lines 13-29).

Introduction:

The term “engineered versions of a virus” is used without clearly defining the basic characteristics of a viral vector. Other herpesviruses that carry foreign genes have been made for medical and veterinary use and these should be included for full understanding. How is latency of these viruses managed?

Response: We thank the reviewer for valuable comments. We have defined the basic characteristic of viral vector in the revised MS (Line 41-78). We provided the example of other HSV-1, adenovirus, lentivirus and AAV as a gene therapy vectors (line 93-116). Moreover, Table 1 listed CMV and other viral vectors (AAV, Ad, HSV-1) based gene therapy. We also provided a link to ongoing viral vector based world-wide clinical trials (1,140). These clinical trials are based on AAV, Ad, and lentiviruses (Lines 108-115).

Properties:

The authors make a bold claim that attenuated Towne and AD169 strains of CMV “cannot replicate in vivo, they grow well in tissue culture, making them well-suited as foundations for viral vectors.”  What evidence exists that the attenuation of these strains results in replication incompetence? The statement seems to be a stretch and most likely incorrect. When administered to CMV naïve human volunteers, Towne strain and AD169 strain, as well as Towne/Toledo chimera viruses were all immunogenic, interpreted as being the result of replication of attenuated virus because inactivated virus was not immunogenic. The section seeks to promote the point that CMV shows a “significant advantage as a vaccine vector” without clearly delineating the reasons why.

Response: We appreciate reviewers’ comment and suggestion. We would like to draw the attention of reviewer toward two studies. AD169 has been evaluated as a live attenuated virus in clinic (Elek and Stern, 1974; Neff et al, 1979), and the results from these studies clearly demonstrated that the virus didn’t cause any active infection in human subjects, with no pathology (symptom) or viral shedding even dosed at 300,000 pfu. This clearly suggests that attenuated AD169 is not able to replicate in vivo (98-101). Potential attenuating mutations have been identified; (Cui et al., 2011) replication-defective virus has been constructed and evaluated in clinical trials; (Wang et al., 2016). The safety of CMV vector could be enhanced by selected genetic modifications. (Marshall et al., 2019).

The immunogenic properties of attenuated vaccine might be from the antigens that expressed in these strains.

Moreover, Merk CMV vaccine in clinical trial is based on AD169 also confirm that AD169 is attenuated and cannot replicate in vivo and below are references.

Wang D, Freed DC, He X, Li F, Tang A, Cox KS, Dubey SA, Cole S, Medi MB, Liu Y, Xu J, Zhang ZQ, Finnefrock AC, Song L, Espeseth AS, Shiver JW, Casimiro DR, Fu TM. A replication-defective human cytomegalovirus vaccine for prevention of congenital infection. Sci Transl Med. 2016 Oct 26;8(362):362ra145. doi: 10.1126/scitranslmed.aaf9387. PMID: 27797961.

Liu Y, Freed DC, Li L, Tang A, Li F, Murray EM, Adler SP, McVoy MA, Rupp RE, Barrett D, Ye X, Zhang N, Beck K, Culp T, Das R, Song L, Vora K, Zhu H, Wang D, Espeseth AS, An Z, Musey L, Fu TM. A Replication-Defective Human Cytomegalovirus Vaccine Elicits Humoral Immune Responses Analogous to Those with Natural Infection. J Virol. 2019 Nov 13;93(23):e00747-19. doi: 10.1128/JVI.00747-19. PMID: 31511385; PMCID: PMC6854503.

Live attenuated Towne strain of HCMV was evaluated in Phase 1 and Phase II clinical trials was found to be safe in human. Below are references that shows the safety of live attenuated Towne strain of HCMV.

  1. Plotkin SA, Farquhar J, Horberger E. 1976.Clinical trials of immunization with the Towne 125 strain of human cytomegalovirus.  Infect. Dis. 134:470–475. 
  2. Plotkin SA, Huang ES. 1985. Cytomegalovirus vaccine virus (Towne strain) does not induce latency.  Infect. Dis.152:395–397.
  3. Plotkin SA, Smiley ML, Friedman HM, Starr SE, Fleisher GR, Wlodaver C, Dafoe DC, Friedman AD, Grossman RA, Barker CF. 1984.Prevention of cytomegalovirus disease by Towne strain live attenuated vaccine. Birth Defects Orig. Artic. Ser. 20:271–287.
  4. Adler SP, Starr SE, Plotkin SA, Hempfling SH, Buis J, Manning ML, Best AM. 1995.Immunity induced by primary human cytomegalovirus infection protects against secondary infection among women of childbearing age.  Infect. Dis.171:26–32

Figure 1 should be replaced by a figure or table that contracts CMV-based vectors with more commonly studied and FDA licensed viruses and vectors based on other herpesviruses as well as on other virus types. The current figure does little to assist the reader in gaining perspective.

Response: We have included a new Table 1 that includes CMV based studies and other viral vector-based gene therapy.

Applications:

This section seeks to summarize what has been gained from studies of surrogate CMV models and so should be more accurately titled. All examples in Figure 2 employ non-human CMVs.  How closely related are these viruses in molecular terms? Can any enter and express vectored genes in human cells or actually replicate in human cells? If so, can they infect humans and be tested as replication competent or nonreplicating vectors? (It appears that RhCMV is actually grown in immortalized human fibroblasts and reports suggest that MCMV can be adapted to human RPE cells. Some overarching perspective is needed here.

Although examples are given in this section, the points are not synthesized or sufficiently integrated.

Response: We have changed the title of the heading to CMV based vector and gene therapy. CMV shows species specificity. HCMV can only productively infect human cells, and MCMV and RhCMV can only productively infect their own host cells. Therefore, MCMV and RhCMV based gene therapy were performed in animals. The knowledge from MCMV and RhCMV models may be translated to HCMV due to their similarities. However, MCMV or RhCMV can enter human cells to express viral proteins and even replicate viral DNA but no infectious viral particles are detected (Tang et al 2006 JVI pmid:16840331; Jurak et al 2006, Embo J, PMID:16688216; Lafemina et al 1989 Viorlogy, PMID:2477948). Therefore, the reviewer’s comments are very important in potential use of animal CMV for developing vector for vaccine and gene therapy, which is included in the revised MS in lines 174-194).

Comments on the Quality of English Language

Very simple 8th grade level text.

Response: English writing is improved.

Reviewer 2 Report

The manuscript by Zeng and colleagues entitled „Exploring the Potential of Cytomegalovirus-Based Vectors: A Review” provides a nicely written and fairly comprehensive overview about the potential of cytomegaloviruses as vaccine vectors. A few suggestions for improvement are listed below.

1.  The authors focus on live-attenuated CMVs as vaccine vectors. These can infect their host and replicate to a limited extent, and they might also enter latency. However, other concepts have been proposed as well, which should be mentioned:

(I) MCMV and RhCMV have been tested as single-cycle vaccines, i.e. viruses that infect the first target cell but do not spread because they lack an essential gene (see PMID 31627457 and references therein). This is a principle similar to MVA (Modified Vaccinia virus Ankara), which also doesn’t spread in humans beyond the first target cell.

(II) The CMVs have also been proposed as non-attenuated transmissible / self-disseminating viruses (PMID 32719452, 35046024) against emerging viruses or for immunocontraception.

2.  RhCMV has been tested extensively as a vaccine vector against Ebola fever (e.g., PMID 21858240, 25820063, 26876974). This is not depicted in Figure 2 and is not mentioned in the manuscript text.

3.  MCMV has been proposed and tested as a vaccine vector for immunocontraception (see papers by Redwood and Shellam). This important aspect is also not mentioned in the manuscript and figure 2.

Minor issues

4.  Line 58. “… will evade additional disease burdens when introduced to humans”. This sentence sounds confusing. Please rephrase.

5.  Lines 278-302. Different font size.

Author Response

Response to Reviewer 2

Comments and Suggestions for Authors

The manuscript by Zeng and colleagues entitled „Exploring the Potential of Cytomegalovirus-Based Vectors: A Review” provides a nicely written and fairly comprehensive overview about the potential of cytomegaloviruses as vaccine vectors. A few suggestions for improvement are listed below.

  1. The authors focus on live-attenuated CMVs as vaccine vectors. These can infect their host and replicate to a limited extent, and they might also enter latency. However, other concepts have been proposed as well, which should be mentioned:

(I) MCMV and RhCMV have been tested as single-cycle vaccines, i.e. viruses that infect the first target cell but do not spread because they lack an essential gene (see PMID 31627457 and references therein). This is a principle similar to MVA (Modified Vaccinia virus Ankara), which also doesn’t spread in humans beyond the first target cell.

Response: We appreciate reviewer’s suggestion. We have included reviewer’s suggestion in the revised MS in lines 497-666).

(II) The CMVs have also been proposed as non-attenuated transmissible / self-disseminating viruses (PMID 32719452, 35046024) against emerging viruses or for immunocontraception.

Response: We included a new paragraph about CMV as a self-disseminating vector to prevent zoonosis in revised MS (Line 667-687).

  1. RhCMV has been tested extensively as a vaccine vector against Ebola fever (e.g., PMID 21858240, 25820063, 26876974). This is not depicted in Figure 2 and is not mentioned in the manuscript text.

 Response: We have included a new paragraph in the revised MS about CMV vectored based EBOV vaccine (line 1181-1194).

  1. MCMV has been proposed and tested as a vaccine vector for immunocontraception (see papers by Redwood and Shellam). This important aspect is also not mentioned in the manuscript and figure 2.

Response: We have included a paragraph on the MCMV vectored immunocontraceptive vaccine in revised MS in line (1196-1209).

Minor issues

  1. Line 58. “… will evade additional disease burdens when introduced to humans”. This sentence sounds confusing. Please rephrase.

Response: We have changed the suggested sentence in revised MS.

  1. Lines 278-302. Different font size.

Response: We thank the reviewer for the careful reading. We have made a suggested change in the revised MS as per suggestion.

Round 2

Reviewer 1 Report

The authors have made substantive corrections that improve the review’s accuracy.